# Extracellular fibrinogen-binding protein released by intracellular *Staphylococcus aureus* suppresses host immunity by targeting TRAF3

Xiaokai Zhang[1,6], Tingrong Xiong[1,6], Lin Gao[1,6], Yu Wang[1,2,6], Luxuan Liu[3], Tian Tian[1], Yun Shi[4], Jinyong Zhang[1], Zhuo Zhao[1], Dongshui Lu[1], Ping Luo[1], Weijun Zhang[1], Ping Cheng[1], Haiming Jing[1], Qiang Gou[1], Hao Zeng[1,7] ✉, Dapeng Yan [5,7] ✉ & Quanming Zou[1,7] ✉

Many pathogens secrete effectors to hijack intracellular signaling regulators in host immune cells to promote pathogenesis. However, the pathogenesis of *Staphylococcus aureus* secretory effectors within host cells is unclear. Here, we report that *Staphylococcus aureus* secretes extracellular fibrinogen-binding protein (Efb) into the cytoplasm of macrophages to suppress host immunity. Mechanistically, RING finger protein 114, a host E3 ligase, mediates K27-linked ubiquitination of Efb at lysine 71, which facilitates the recruitment of tumor necrosis factor receptor associated factor (TRAF) 3. The binding of Efb to TRAF3 disrupts the formation of the TRAF3/TRAF2/cIAP1 (cellular-inhibitor-of-apoptosis-1) complex, which mediates K48-ubiquitination of TRAF3 to promote degradation, resulting in suppression of the inflammatory signaling cascade. Additionally, the Efb K71R mutant loses the ability to inhibit inflammation and exhibits decreased pathogenicity. Therefore, our findings identify an unrecognized mechanism of *Staphylococcus aureus* to suppress host defense, which may be a promising target for developing effective anti-*Staphylococcus aureus* immunomodulators.

*Staphylococcus aureus* (*S. aureus*) is one of the most common bacterial strains causing hospital- and community-acquired pneumonia associated with significant mortality worldwide[1,2]. It is estimated that ~85% of the human population carries or previously carried *S. aureus*[3]. The nasal cavity is the main site of *S. aureus* colonization in the human body[4], providing a pathogen reservoir that significantly increases the chance of secondary *S. aureus* pulmonary infection. However, the morbidity associated with *S. aureus* pneumonia is far below the bacterial carrying rate due to the powerful host immune system. Resident macrophages, an important class of innate immune cells, are the first to encounter invading pathogens[5]. Besides killing the invading pathogens, macrophages can produce pro-inflammatory cytokines,

[1]National Engineering Research Center of Immunological Products, Department of Microbiology and Biochemical Pharmacy, College of Pharmacy, Third Military Medical University, Chongqing 400038, China. [2]Department of Basic Courses, NCO School, Third Military Medical University, Shijiazhuang 050081, China. [3]College of Medicine, Southwest Jiaotong University, Chengdu 610083, China. [4]Institute of Biopharmaceutical Research, West China Hospital, Sichuan University, Chengdu, Sichuan 610041, China. [5]Department of Immunology, School of Basic Medical Sciences, Shanghai Institute of Infectious Disease and Biosecurity & Shanghai Public Health Clinical Center, Fudan University, Shanghai 200032, China. [6]These authors contributed equally: Xiaokai Zhang, Tingrong Xiong, Lin Gao, Yu Wang. [7]These authors jointly supervised this work: Hao Zeng, Dapeng Yan, Quanming Zou. ✉e-mail: zeng1109@163.com; dapengyan@fudan.edu.cn; qmzou2007@163.com

chemokines, and lipid mediators that recruit other innate immune cells, such as neutrophils, monocytes, and dendritic cells, to orchestrate immune responses and fight infections[6]. Therefore, resident macrophages represent a critical defense line that *S. aureus* must overcome in order to propagate in the host[7].

*S. aureus* secretes multiple effectors that functionally evade or inhibit host immune responses[8]. Previous pathogenic studies on *S. aureus* have mainly focused on the effect of its virulent factors on the cytomembrane and its receptors in innate immune cells. Several hemolysins, leukocidins, and phenol-soluble modulins have been identified to lyse cells by forming pores on the membranes of macrophages and other innate immune cells[9]. Other bacterial factors, including staphylococcal superantigen-like protein 3 and lipoylated E2 subunit of the pyruvate dehydrogenase complex, released from *S. aureus* can suppress macrophage activation through inhibiting Toll-like receptors (TLR) activation[10,11]. Recently, it has been demonstrated that *S. aureus* can invade and survive within macrophages and other host cells[12]. Even in a hostile environment, such as that in macrophages, *S. aureus* can develop specific countermeasures to evade the immune response. Numerous intracellular bacteria have been shown to weaken host immune defenses by secreting virulent factors capable of hijacking macrophage signaling pathways[13]. However, which effectors are secreted by intracellular *S. aureus* to manipulate the signaling pathways of macrophages and their underlying mechanisms remains unclear.

To inhibit the intracellular survival of invading pathogens, the inflammatory signaling pathway promotes the expression of various pro-inflammatory cytokines including tumor necrosis factor (TNF), interleukin 1β (IL-1β), IL-6, and IL-12[14,15]. In the present study, we found that extracellular fibrinogen-binding protein (Efb) released by intracellular *S. aureus* inhibits pro-inflammatory cytokine expression in macrophages. Previous studies have suggested that Efb can block the function of C3b, inhibit the formation of platelet-leukocyte complexes, and bind fibrinogen to prevent neutrophil activation[16,17]. However, little is known about the role of intracellular Efb in modulating host inflammatory signaling pathways.

In this work, we demonstrate that intracellular *S. aureus* secretes Efb into the cytoplasm of macrophages to inhibit expression of pro-inflammatory cytokines and suppress host immunity by interacting with tumor necrosis-associated factor 3 (TRAF3). Mechanistically, the interaction between Efb and TRAF3 requires K27-linked ubiquitination of Efb mediated by a host E3 ubiquitin ligase, RING finger protein 114 (RNF114).

## Results

### Efb inhibits host pro-inflammatory responses

To identify the anti-inflammatory components secreted by intracellular *S. aureus*, we tested mature chains of 78 secretory proteins of *S. aureus* on nuclear factor-κB (NF-κB) activation in HEK293T cells using a dual-luciferase reporter gene assay (Supplementary Fig. 1a, b). Efb was one of the proteins shown to inhibit NF-κB activation induced by TNF-α (Supplementary Fig. 1b). Efb was co-expressed with an NF-κB reporting gene in HEK293T cells to verify the inhibitory role of intracellularly expressed Efb on activation of the NF-κB pathway (Supplementary Fig. 1c). Next, we found that adenoviral vector-mediated intracellular overexpression of Efb strongly suppressed phosphorylation of p65 (Supplementary Fig. 1d) and mRNA levels of pro-inflammatory cytokines, including *Tnf*, *Il1β*, *Il6*, and *Il12p40* (Supplementary Fig. 1e), in peritoneal macrophages (PMs) stimulated with heat-killed *S. aureus*. These results suggest that intracellular Efb inhibits the transcription of pro-inflammatory cytokines likely through inhibiting the NF-κB pathway.

To validate the anti-inflammatory activity of physiological Efb in an infection model, we generated an *S. aureus* Newman strain with an Efb-deletional mutant (ΔEfb) and a complementation strain (ΔEfb +

Flag − Efb). As shown in Supplementary Fig. 2a, western blot analysis with anti-Efb or anti-Flag antibodies and silver staining of polyacrylamide gel electrophoresis (PAGE) revealed Efb expression in the culture supernatant of the corresponding strains. Consistent with a previous report[18], the absence of Efb did not affect *S. aureus* growth (Supplementary Fig. 2b).

Next, we found that macrophages infected with the ΔEfb strain significantly increased the mRNA levels of *Tnf*, *Il1β*, *Il6*, and *Il12p40* in infected macrophages, while the ΔEfb + Flag − Efb strain restored the inhibitory effect of Efb (Fig. 1a, Supplementary Fig. 3a). However, the inhibitory effect of Efb on the expression of inflammatory cytokines was lost after stimulation with only the culture supernatants of *S. aureus* (Supplementary Fig. 3b) or non-contact co-culture of macrophages and *S. aureus* in Transwells (Supplementary Fig. 3c). These data suggest that Efb released by intracellular *S. aureus*, not Efb released by extracellular *S. aureus*, exerts anti-inflammatory functions. In the control experiments, we found no difference in intracellular *S. aureus* loading among macrophages infected with Newman, ΔEfb, or ΔEfb + Flag − Efb strains for 6 h (Supplementary Fig. 3d, e, GFP did not affect the expression of Efb in *S. aureus*; Supplementary Fig. 3f), indicating that the above-observed differences in mRNA levels of inflammatory cytokines are Efb-dependent. Compared to the Newman strain, the ΔEfb strain also exhibited accelerated phosphorylation of p65, p38, JNK, and ERK in macrophages (Supplementary Fig. 4a, b), suggesting that Efb may also inhibit the mitogen-activated protein kinase (MAPK) pathway in addition to NF-κB pathway. These results provide further evidence that physiological Efb released by intracellular *S. aureus* suppresses *S. aureus*-triggered inflammatory responses in macrophages.

To investigate the role of Efb in vivo, we established a trachea cannula infection model by challenging 6-week-old C57BL/6 mice with Newman, ΔEfb, or ΔEfb + Flag − Efb strains. At 24 h post infection, there was increased expression of TNF-α, IL-1β, IL-6, and IL-12 in the lungs of mice infected with the ΔEfb strain compared to those infected with the Newman and ΔEfb + Flag − Efb strains (Fig. 1b). In agreement, the bacterial burden in the lungs of Newman and ΔEfb + Flag − Efb infected mice was significantly higher than that of ΔEfb infected mice (Fig. 1c). There were also more intact alveolar spaces, along with less infiltration of neutrophils and lymphocytes, in the lungs of ΔEfb infected mice (Fig. 1d, e). We also found that ΔEfb directly led to an increase in survival rates in lethal pneumonia (Supplementary Fig. 5a) and bacteremia (Supplementary Fig. 5b) models, as well as alleviated festering areas in the skin infection model (Supplementary Fig. 5c, d), compared to Newman and ΔEfb + Flag − Efb infected mice. Together, these results suggest that Efb released by intracellular *S. aureus* may act as an inhibitor of host inflammatory responses, making Efb an important factor for *S. aureus* infection.

### Efb interacts with TRAF3

Immunoelectronmicroscopy and immunofluorescent confocal microscopy showed that Efb was secreted by intracellular *S. aureus* into the cytoplasm of the macrophages and gradually accumulated over time during infection (Supplementary Fig. 6a, b). TLR signaling, especially TLR2, plays an important role in detecting *S. aureus* and inducing the expression of pro-inflammatory cytokines during infection[10,19]. Therefore, using a co-immunoprecipitation assay, we screened Efb-interactive proteins from a list of key signaling molecules in the TLR pathway. TRAF3 appeared to be the only Efb-associated protein (Supplementary Fig. 7a and Fig. 2a, b). Further experiments in ΔEfb + Flag − Efb infected macrophages demonstrated that Efb released by intracellular *S. aureus* interacted physiologically with TRAF3 (Fig. 2c, d and Supplementary Fig. 7b). The in vitro GST-pull down assay also demonstrated a direct interaction between purified Efb and TRAF3 (Supplementary Fig. 7c, d). In addition, the zinc finger domain of TRAF3 was responsible for the interaction (Fig. 2e, f).

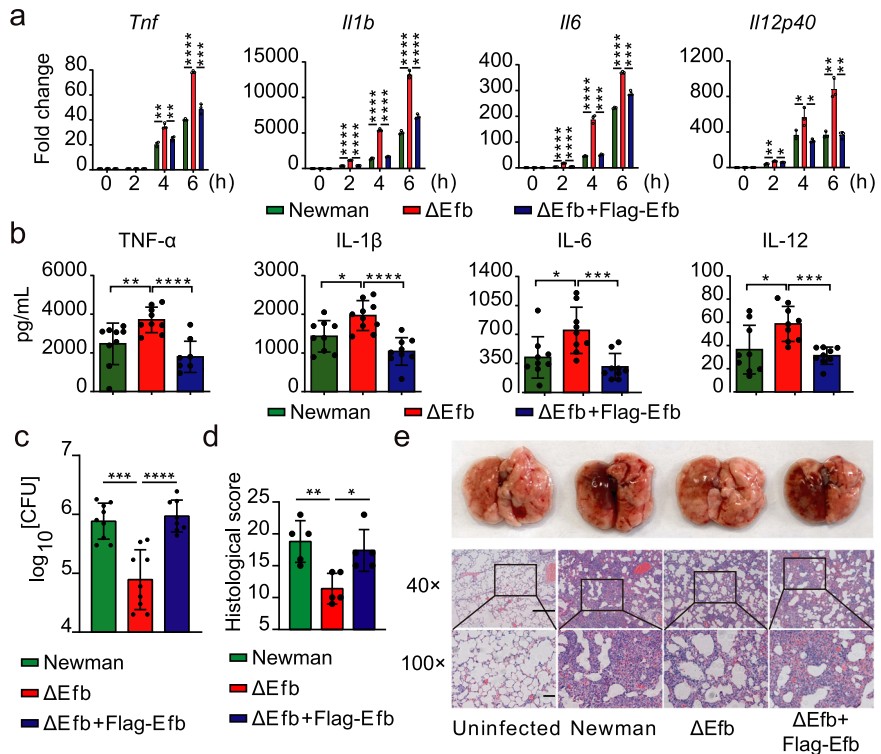

**Fig. 1 | Efb inhibits host pro-inflammatory responses. a** Quantitative polymerase chain reaction (qPCR) analysis of *Tnf*, *Il1β*, *Il6*, and *Il12p40* mRNA from alveolar macrophage (MH-S) infected with Newman, ΔEfb, or ΔEfb + Flag − Efb for indicated times (MOI = 25; \*\*$P$ = 0.0014, 0.0052, \*\*\*\*$P$ < 0.0001, \*\*\*$P$ = 0.0003 in sequence, *Tnf*; \*\*\*\*$P$ < 0.0001, *Il1β*; \*\*\*\*$P$ < 0.0001, <0.0001, <0.0001, \*\*\*$P$ = 0.0001, \*\*\*\*$P$ < 0.0001, \*\*\*$P$ = 0.0002, in sequence, *Il6*; \*\*$P$ = 0.0019, \*$P$ = 0.0126, 0.0391, 0.0118, \*\*$P$ = 0.0020, 0.0020, in sequence, *Il12p40*). **b** ELISA quantification of TNF-α, IL-1β, IL-6, and IL-12 levels in lung tissue homogenized in 1 ml PBS 24 h after infection with Newman, ΔEfb, or ΔEfb + Flag − Efb. C57BL/6 mice were infected by intratracheal administration with the test strains (2 × $10^8$ CFUs per mouse) for the indicated times (\*\*$P$ = 0.0095, \*\*\*\*$P$ < 0.0001, in sequence, TNF-α; \*$P$ = 0.0109,

\*\*\*\*$P$ < 0.0001, in sequence, IL-1β; \*$P$ = 0.0187, \*\*\*$P$ = 0.0008, in sequence, IL-6; \*$P$ = 0.0206, \*\*\*$P$ = 0.0002, in sequence, IL-12). **c** Quantification of the bacterial CFUs of lung tissue homogenates obtained in b (\*\*\*$P$ = 0.0001, \*\*\*\*$P$ < 0.0001). **d**, **e** Histopathology of lung tissues was assessed in H&E sections stained from mice infected for 24 h; scale bars, 1,000 μm (top) and 200 μm (bottom), the boxed areas at the top are enlarged below (\*\*$P$ = 0.0036, \*$P$ = 0.0110). Student's two-tailed unpaired *t*-test (**a**, **b**, **d**) or two-tailed Mann−Whitney *U* test (**c**) was used for statistical analysis. Data are representative of three experiments with at least three independent biological replicates. The bars show the mean and standard deviation of $n$ = 3 (**a**), $n$ = 9 (**b**, **c**), and $n$ = 5 (**d**) per group. Source data are provided as a Source Data file.

## Efb inhibits pro-inflammatory responses by stabilizing TRAF3

Next, we examined whether TRAF3 mediated the inhibition effects of Efb on host pro-inflammatory responses. We generated conditional knockout mice for TRAF3 (genotype: TRAF3[flox/flox, Lyz2-Cre]) using CRISPR/Cas9-mediated genome editing and isolating the PMs from these mice. TRAF3 deficiency effectively abrogated Efb's inhibitory effects on the mRNA levels of *Tnf*, *Il1β*, *Il6*, and *Il12p40* in PMs (Fig. 3a). Furthermore, Efb's inhibitory effects were lost in lungs of TRAF3[flox/flox, Lyz2-Cre] mice, but not in TRAF3[flox/flox] mice (Fig. 3b). Consistent with this result, there were no significant differences in the pulmonary bacteria burdens and histopathology between TRAF3[flox/flox, Lyz2-Cre] mice infected with the Newman and ΔEfb strains (Fig. 3c−e). TRAF3 mediates NF-κB, MAPK, and type I interferon pathways[20]. Therefore, we used NIK SMI1 (NIK inhibitor), GSK8612 (TBK1 inhibitor), and 5Z-7-oxozeaeno (TAK1 inhibitor) to test which pathway mainly mediates pro-inflammatory cytokine production of macrophages infected with *S. aureus*. The results show that 5Z-7-oxozeaeno, but not NIK SMI1 and GSK8612, inhibited pro-inflammatory cytokine production. TAK1 mainly mediated the canonical NF-κB and MAPK pathways[21]. Therefore, the canonical NF-κB and MAPK pathways may play a dominant role in this mechanism. And the anti-inflammatory effects of Efb could still be observed in macrophages treated with NIK SMI1 and GSK8612, but not with 5Z-7-oxozeaeno (Supplementary Fig. 8a−c). Collectively, these in vitro and in vivo results indicate that Efb may suppress host inflammatory responses via inhibiting TRAF3-mediated canonical NF-κB and MAPK pathway activation.

Previous studies have demonstrated that K48 and K63 ubiquitination of TRAF3 is essential for activation of the NF-κB and MAPK pathways[22–25]. Here, we found that, compared to the Newman strain, the ΔEfb strain markedly reduced the amounts of TRAF3 in macrophages, suggesting that Efb may stabilize TRAF3 (Fig. 4a and Supplementary Fig. 9a). Consistently, both overexpressed and physiological Efb significantly reduced the K48-linked ubiquitination of TRAF3 in HEK293T cells and macrophages (Fig. 4b, c, Supplementary Fig. 9b). However, we found little evidence of K63-linked ubiquitination of TRAF3 in macrophages infected with *S. aureus* (Supplementary Fig. 9c), indicating that Efb may mainly affect the K48-linked ubiquitination of TRAF3. The degradative ubiquitination of TRAF3 is typically mediated by cIAP1/2, and the modification effects require TRAF2 to act as a bridge between TRAF3 and cIAP1/2[26,27]. We found that TRAF2-mediated K48-linked ubiquitination of TRAF3 in macrophages infected by *S. aureus* (Supplementary Fig. 9d), and Efb could prevent TRAF3 from interacting with TRAF2 and cIAP1 (Fig. 4d, e, Supplementary Fig. 9e), resulting in inhibition of K48-linked ubiquitination of TRAF3 in both a cell-free ubiquitylation reaction system and HEK293T cells (Fig. 4f, Supplementary Fig. 9f). We further demonstrated that the zinc finger domain of TRAF3 was responsible for its binding to TRAF2 (Supplementary Fig. 9g), suggesting that Efb could competitively bind TRAF3 and affect TRAF2 binding to TRAF3.

Lysines 106 and 155 of TRAF3 were reported as the main K48-linked ubiquitination sites[28], and our results showed that Efb mainly inhibited the K48-linked ubiquitination of lysine 155 in the zinc finger

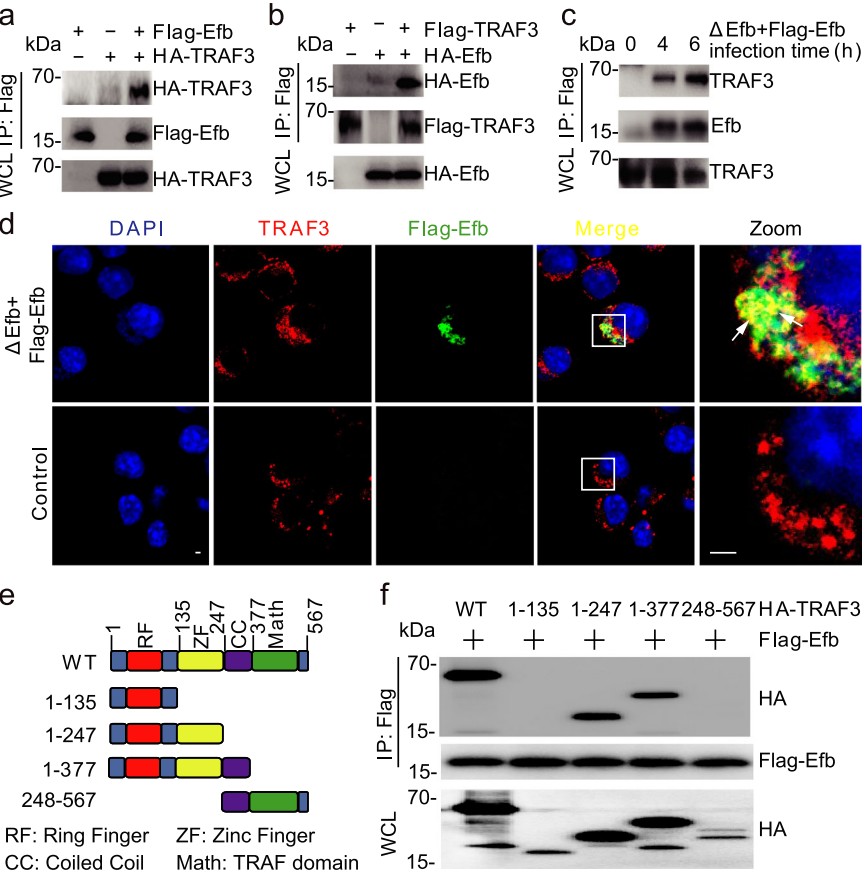

**Fig. 2 | Efb interacts with TRAF3. a, b** Immunoblots of whole cell lysates (WCL) and immunoprecipitation (IP) products of WCL from HEK293T cells transfected with indicated plasmids. **c** Immunoblots of WCL and IP products of WCL from MH-S infected with ΔEfb + Flag − Efb for indicated times (MOI = 25). **d** Immunofluorescence assay of MH-S infected with ΔEfb + Flag − Efb for 4 h (MOI = 25), scale bars, 2 μm. **e** Truncation of TRAF3. **f** Immunoblots of WCL and IP products of WCL from HEK293T cells transfected with indicated plasmids; WT, full-length TRAF3. Data are representative of three experiments with at least three independent biological replicates. Source data are provided as a Source Data file.

domain of TRAF3 (Fig. 4g and Supplementary Fig. 9h). Next, we expressed wild-type TRAF3 in TRAF3 knockout macrophages to restore the function of TRAF3 and expressed TRAF3 K155R in TRAF3 knockout macrophages to mimic the effects of Efb on TRAF3. After infection, the pro-inflammatory cytokines induced by the ΔEfb stain remained higher than the Newman strain in wild-type TRAF3, but not in TRAF3 K155R-expressing macrophages (Supplementary Fig. 9i). Taken together, these results suggest that Efb disturbs the TRAF3/TRAF2/cIAP1 complex, inhibits the K48-linked ubiquitination of TRAF3 at lysine 155, and prevents TRAF3 degradation in macrophages during *S. aureus* infections.

### RNF114 mediated K27-linked ubiquitination of Efb

Several studies have suggested that bacteria can utilize the post-translational modification system of the host to modify their effectors for pathogenesis[29,30]. Given that Efb is small in size (mature protein, 15kD) and contains 20 lysine residues that are common sites for ubiquitination, we investigated the ubiquitination of intracellular Efb. Plasmids expressing different ubiquitins (Ub, K6, K11, K27, K29, K33, K48, and K63) were co-transfected with Efb into HEK293T cells. Co-immunoprecipitation results revealed multiple types of ubiquitination of Efb, with K27-linked polyubiquitination being most dominant (Supplementary Fig. 10a). Furthermore, we detected the endogenous K27-linked polyubiquitin conjugates of intracellular Efb in macrophages infected with the ΔEfb + Flag − Efb strain (Fig. 5a, Supplementary Fig. 10b), indicating that Efb is polyubiquitinated by K27 in host cells. We also carried out proteomics analysis on immunoprecipitated

overexpressed Efb in HEK293T cells using tandem mass spectrometry (MS/MS) in Q ExactiveTM Plus (Thermo) coupled online to UPLC (Supplementary Table 3). We identified a protein named RNF114 as an E3 ligase with K27-linked polyubiquitination activity[31]. Overexpression co-immunoprecipitation (Fig. 5b, c) and endogenous co-immunoprecipitation (Fig. 5d, Supplementary Fig. 10c) consistently demonstrated reproducible Efb interaction with RNF114. Over-expression of RNF114 also increased the K27-linked (but not other types), polyubiquitination of Efb in HEK293T cells (Fig. 5e, Supplementary Fig. 11), and RNF114 KD by specific siRNA reduced the K27-linked polyubiquitination of Efb in macrophages (Fig. 5f). These results collectively suggest that RNF114 is an important ubiquitinase that mediates K27-linked polyubiquitination of Efb.

To determine which lysine residue of Efb was modified by K27-linked polyubiquitination, we constructed 20 Efb mutants by replacing lysine with arginine and co-transfected these mutants with a plasmid expressing K27 ubiquitin in HEK293T cells. The immunoprecipitation results revealed that the K71R mutant significantly reduced the K27-linked polyubiquitination of Efb (Supplementary Fig. 12a). Consistently, RNF114 did not promote the K27-linked ubiquitination of K71R mutant of Efb in HEK293T cells (Fig. 5g). Next, we replaced all other Efb lysines with arginines, except lysine 71, and RNF114 still maintained the K27-linked polyubiquitination of Efb at lysine 71 (Supplementary Fig. 12b, c). Moreover, physiological K27-linked ubiquitination of the K71R mutant of Efb disappeared in macrophages (Fig. 5h). These results suggest that RNF114 can mediate K27-linked polyubiquitinion of Efb at lysine 71.

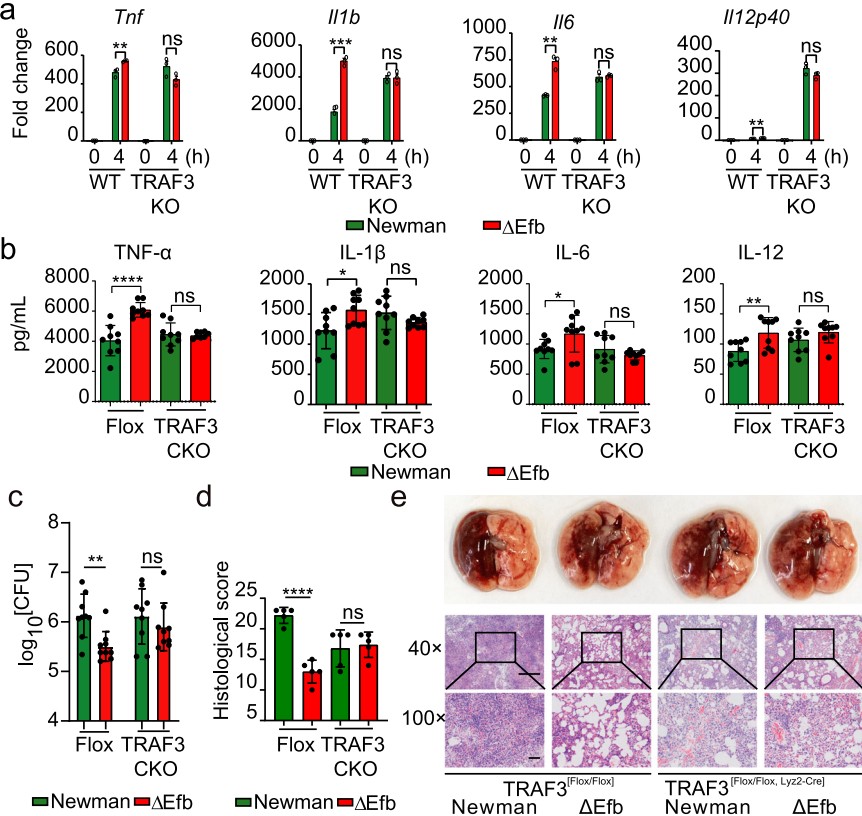

**Fig. 3 | Efb inhibits host pro-inflammatory responses via TRAF3. a** qPCR analysis of *Tnf*, *Il1b*, *Il6*, and *Il12p40* mRNA of PMs infected with Newman, ΔEfb (MOI = 25) for 4 h; wild-type PMs (WT) were isolated from TRAF3[flox/flox] mice; TRAF3 knockout (KO) PMs were isolated from TRAF3[flox/flox, Lyz2-Cre] mice (**P = 0.0096, *Tnf*; ***P = 0.0001, *Il1b*; **P = 0.0011, *Il6*; **P = 0.0015, *Il12p40*). **b** ELISA quantification of TNF-α, IL-1β, IL-6, and IL-12 levels in lung tissues from TRAF3[flox/flox] (Flox) or TRAF3[flox/flox, Lyz2-Cre] (TRAF3 KO) mice, homogenized in 1 ml PBS and infected with Newman, ΔEfb for 24 h (2 × 10^8 CFUs per mouse; ****P < 0.0001, TNF-α; *P = 0.0184, IL-1β; *P = 0.0427, IL-6; **P = 0.0089, IL-12). **c** Quantification of the bacterial CFUs of

lung tissue homogenates obtained in **b** (**P = 0.0030). **d**, **e** Histopathology of lung tissues was assessed in H&E sections stained from mice infected for 24 h; scale bars, 1000 μm (top) and 200 μm (bottom), the boxed areas at the top are enlarged below (****P < 0.0001). Student's two-tailed unpaired *t*-test (**a**, **b**, **d**) or two-tailed Mann–Whitney *U* test (**c**) was used for statistical analysis. Data are representative of three experiments with at least three independent biological replicates. The bars show the mean and standard deviation of n = 3 (**a**), n = 9 (**b**, **d**), and n = 5 (**e**) mice per group. Source data are provided as a Source Data file.

## Ubiquitination of Efb facilitates the stabilization of TRAF3

To investigate whether RNF114 is involved in suppressing TRAF3 signaling by Efb, we observed the enhancing effect of RNF114 on Efb inhibiting K48-linked ubiquitination of TRAF3 (Supplementary Fig. 13a). When compared to wild-type Efb, the K71R mutant appeared to reverse the inhibition of Efb on the K48-linked ubiquitination of TRAF3 (Supplementary Fig. 13b). The results obtained from ΔEfb + Flag − Efb and ΔEfb + Flag − Efb K71R infected macrophages (Supplementary Fig. 13c–e) showed that the K71R-Efb mutant reduced the inhibitory effects on TRAF2 and cIAP1 mediated K48-linked poly-ubiquitination of TRAF3 at physiological levels. Furthermore, we found that ΔEfb infection in RNF114 KD macrophages resulted in an even lower level of pro-inflammatory cytokine expression compared to Newman infection (Supplementary Fig. 13f), and we found that RNF114 affected the TRAF3 expression level only when Efb was present (Supplementary Fig. 13g), suggesting that RNF114 is involved in the regulation of inflammatory cytokine production during *S. aureus* infection in macrophages independent of Efb and TRAF3. Together, these results suggest that RNF114 promotes K27-linked ubiquitination of Efb to facilitate TRAF3 stabilization.

## Ubiquitinated Efb inhibits host pro-inflammatory responses

Next, the reconstitution experiments using ΔEfb infected macrophages demonstrated that wild-type Efb, but not the K71R-mutant strain, was able to restore inhibition of infection-induced pro-inflammatory cytokine production (Fig. 6a, Supplementary Fig. 13c). To

investigate the function of the K71R mutant on the innate immune response in vivo, we infected 6-week-old mice with the Newman, ΔEfb, ΔEfb + Flag − Efb, or ΔEfb + Flag − Efb K71R strains and evaluated the expression levels of pro-inflammatory cytokines in the lungs. Compared with ΔEfb infection, ΔEfb + Flag − Efb, but not ΔEfb + Flag − Efb (K71R), significantly suppressed pro-inflammatory cytokine production in the affected lungs (Fig. 6b). The bacterial load of ΔEfb and ΔEfb + Flag − Efb (K71R) strains in the lungs was significantly lower than that of the Newman and ΔEfb + Flag − Efb strains (Fig. 6c). Consistently, histopathology results revealed more severe lung damage and more neutrophil and lymphocyte infiltration in the lungs of ΔEfb mice reconstituted with wild-type but not the K71R-mutant Efb strains (Fig. 6d, e). These results suggest that the pathogenesis of *S. aureus* might rely on the K27-linked ubiquitination of Efb at its lysine 71 residue.

## Discussion

Opsonophagocytic killing conducted by macrophages and neutrophils is critical for the host to combat *S. aureus*, a classical extracellular pathogen[7]. However, in recent years, accumulating reports revealed that *S. aureus* can invade and survive within macrophages, neutrophils, and other host cells[12]. As a reservoir of *S. aureus*, macrophages play an important role in regulating the inflammatory responses that are critical for eliminating invading pathogens[6,12]. To achieve a successful infection, *S. aureus* must overcome activation of the inflammatory pathways of macrophages.

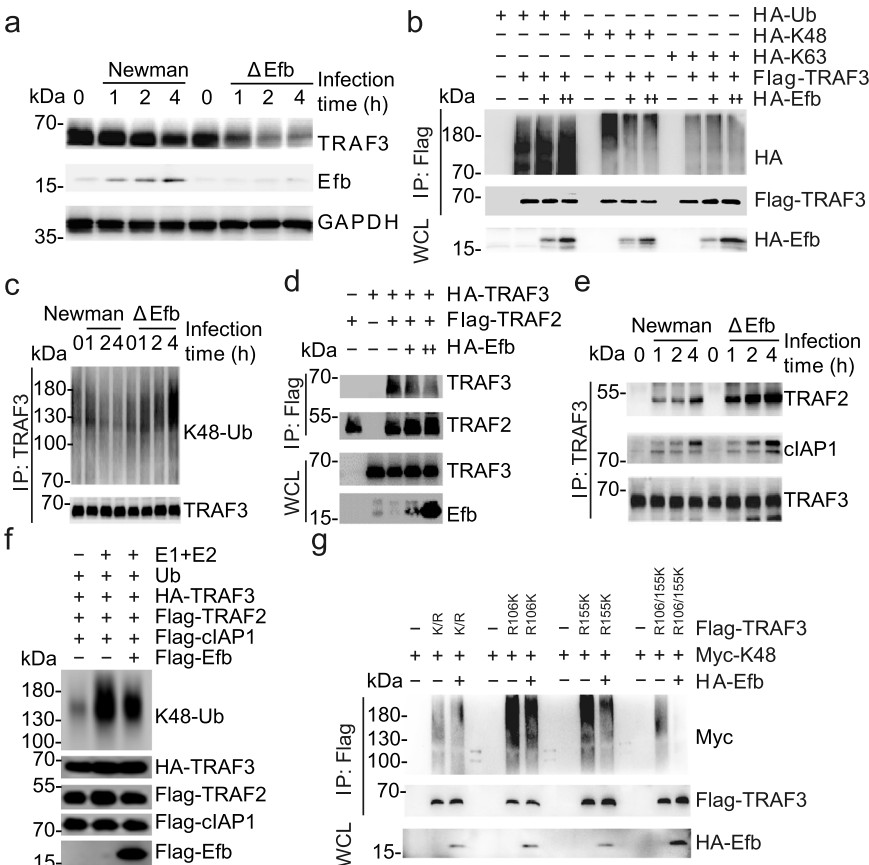

**Fig. 4 | Efb stabilizes TRAF3. a** Immunoblots of WCL from MH-S infected with Newman and ΔEfb (MOI = 25). **b** Immunoblots of WCL and IP products of WCL from HEK293T cells transfected with indicated plasmids. Ub, wild-type ubiquitin; K48, Ub with a single 48 lysine residue left; K63 Ub with a single 63 lysine residue left. **c** Immunoblots of IP products of WCL from MH-S infected with Newman and ΔEfb (MOI = 25). **d** Immunoblots of WCL and IP products from HEK293T cells transfected with indicated plasmids. **e** Immunoblots of IP products from MH-S infected with Newman and ΔEfb (MOI = 25). **f** Efb inhibits K48-linked ubiquitin chains on TRAF3 in vitro. **g** Immunoblots of WCL and IP products from HEK293T cells transfected with indicated plasmids. K/R, replace all lysine of TRAF3 with arginine; R106K, replace 106 arginine of K/R with lysine; R155K, replace 155 arginine of K/R with lysine. R106/155K, replace 106 and 155 arginine of K/R with lysine. Data are representative of three experiments with at least three independent biological replicates. Source data are provided as a Source Data file.

Secreting multiple effectors is a unique strategy that *S. aureus* has adopted to cope with environmental challenges[9]. However, whether and how the secreted effectors of intracellular *S. aureus* interfere with the inflammatory pathways of macrophages remains unclear. In the present study, we screened 78 secreted effectors on activation of the NF-κB pathway. The results showed that more than half of the secretory proteins decreased NF-kB activation. We conducted several bioinformatics and literature searches on these proteins, especially α-hemolysin (WP-000857483.1), serine protease SplE (WP-001038759.1), and cysteine protease staphopain A (WP-000827748.1). We found that the inhibitory effects of these three proteins were false positives due to the fact that overexpressing these proteins in HEK293T cells led to cell deformation or death (Supplementary Fig. 1f). Moreover, this validation work is still in progress. Efb was the first protein we discovered that could inhibit both NF-κB and MAPK, as well as inhibit *S. aureus*-induced expression of pro-inflammatory cytokines, including TNFα, IL-1β, IL-6, and IL-12, in vitro and in vivo, respectively. And we determined that the Efb does not influence the in vitro killing ability of isolated macrophages (Supplementary Fig. 3d–f) or neutrophils (Supplementary Fig. 14) in vitro. In the early stages of infection, increased levels of TNFα and IL-1β are critical for eliminating bacterium in human *S. aureus* bacteremia[32]. IL-1β has also been shown to be a key cytokine for the eradication of *S. aureus* in experimental models[33]. Our results indicate that decreased bacterial burdens in lung tissues infected with ΔEfb could be attributed to elevated TNF-α and IL-1β levels,

which may lead to more neutrophil or other immune cell chemotaxis or activation. Thus, inhibiting host pro-inflammatory responses may be one way by which Efb contributes to the pathogenicity of *S. auraues*.

Although the post-translational modification system in bacteria is very simple in comparison to eukaryotes[34], accumulating studies have shown that some bacteria effectors can be modified by the host post-translational modification system[29]. Under certain conditions, some bacteria effectors can disturb host post-translational modification[35] to facilitate infection. Our present study found that a host E3 ligase, RNF114, ubiquitinates Efb of *S. aureus* to suppress host immunity. K27-linked ubiquitination serves a variety of functions, including enhancing protein–protein interactions, promoting proteasomal degradation, and providing binding platforms for DNA repair proteins[36]. RNF114 has been reported as a host E3 ligase with K27-linked polyubiquitination activity with degradation effects in porcine and sea perch[37,38]. However, our results showed that K27-linked ubiquitination of Efb by RNF114 did not lead to degradation of Efb but promoted the interaction between Efb and TRAF3.

Efb has been identified as an immune evasion effector, and its complement inhibitory effect depends on its C-terminal[16]. However, our results demonstrate that replacing lysine 71 with arginine in the N-terminal of Efb can eliminate its inhibitory effects in vivo and in vitro, suggesting that the inhibitory effects of Efb are independent of its complement effects in vivo. On the other hand, the immunosuppressive abilities of Efb were shown to be correlated with inhibiting the

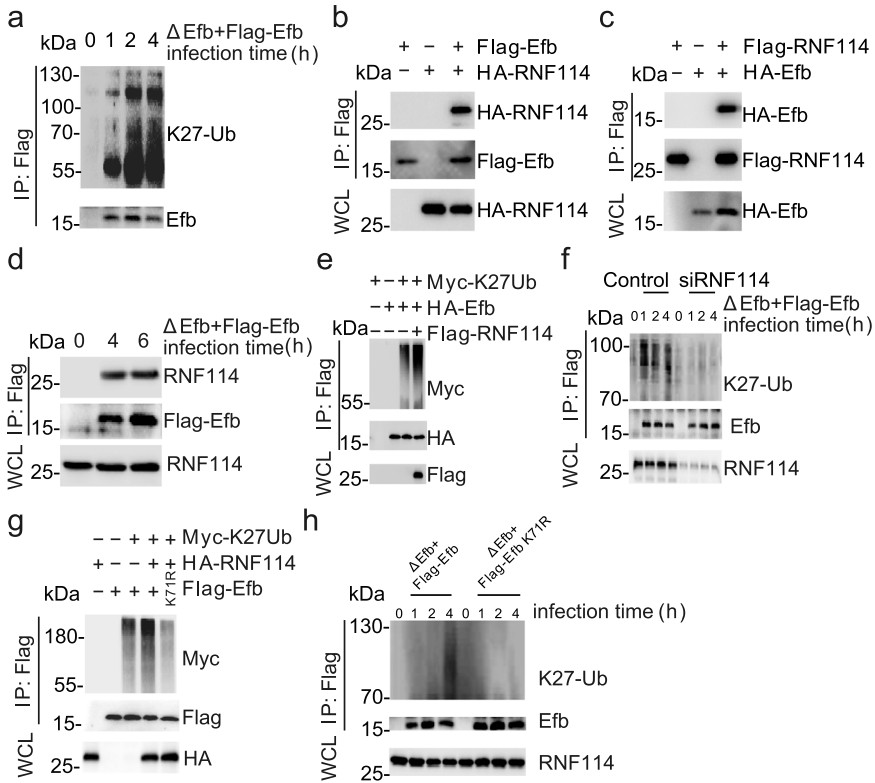

**Fig. 5 | Efb is K27-ubiquitinated at K71 by host RNF114. a** Immunoblots of IP products of WCL from MH-S infected with ΔEfb + Flag − Efb for indicated times (MOI = 2 5). **b, c** Immunoblots of WCL and IP products from WCL of HEK 293 T cells transfected with indicated plasmids. **d** Immunoblots of IP products of WCL from MH-S infected with ΔEfb + Flag − Efb for indicated times (MOI = 25). **e** Immunoblots of WCL and IP products from HEK293T cells transfected with indicated plasmids. **f** Immunoblots of WCL and IP products from MH-S or RNF114 knockdown (KD) MH-S infected with ΔEfb + Flag − Efb for indicated times (MOI = 25). **g** Immunoblots of WCL and IP products from HEK293T cells transfected with indicated plasmids. K71R, replace 71 lysine of Efb with arginine. **h** Immunoblots of WCL and IP products from MH-S infected with ΔEfb + Flag − Efb or ΔEfb + Flag − Efb K71R for indicated times (MOI = 25). Data are representative of three experiments with at least three independent biological replicates. Source data are provided as a Source Data file.

formation of platelet-leukocyte complexes and binding fibrinogen to prevent neutrophil activation, which depends on the N-terminal of Efb[17]. However, lysine 71 of Efb was not the key residue in the interaction between Efb and fibrinogen according to a previous report[39]. Platelets have been reported to play an important role in inflammation[40]. However, there is no detailed information as to which amino acid of Efb plays a key role in the interaction between the Efb N-terminal and platelets. Our results demonstrate that the direct inhibitory effect of Efb on pro-inflammatory cytokines was accomplished through interaction with TRAF3 in myeloid cells, including mature macrophages, monocytes, and granulocytes. The anti-inflammatory effects of Efb are critical to the in vivo change in *S. aureus* load and pathogenicity in mice. However, it is still possible that our in vivo results were partly due to the interaction between Efb and platelets or other unknown mechanisms.

TRAF3 is a tri-faced immune regulator that has distinct roles depending on its targeted receptors, even within the same cell, and is also highly cell-type-dependent[20]. TRAF3 has been reported to positively regulate type I interferon production while negatively regulating NF-κB and MAPK pathways[22]. Our results demonstrate that TRAF3 regulates *S. aureus*-induced activation of NF-κB and MAPK pathways in macrophages. *S. aureus* infection induces the degradation of TRAF3 in macrophages. However, Efb can inhibit degradation by disturbing the K48-linked ubiquitination of TRAF3 in HEK293T cells, PMs, and alveolar macrophages (MH-S). K48-linked ubiquitination of TRAF3 for degradation is accomplished by cIAP1/2, which requires the bridging effects of TRAF2. Our data indicate that the interaction between TRAF3 and TRAF2 was disturbed by Efb, and the inhibition on K48-linked

ubiquitination of TRAF3 by Efb was accomplished by disrupting the TRAF3-TRAF2-cIAP1/2 complex.

As stated in previous reports, TLR2 and MyD88 in innate immune cells play a key role in detecting *S. aureus*[41-43]. We demonstrated that TLR2 and MyD88 were critical in promoting pro-inflammatory responses of macrophages infected by *S. aureus* (Supplementary Fig. 15a). As stated in a report by Perkins et al.[44], macrophages stimulated by P3C, a TLR2 ligand, overnight (≥24 h) upregulated TRAF3 protein expression. However, their results also showed that the TRAF3 mRNA levels were not significantly increased until 8 h post treatment. In the present study, we measured TRAF3 protein expression immediately after *S. aureus* infection using wild-type, TLR2−/−, and MyD88−/− macrophages. We found that the decrease in TRAF3 protein expression was related to the TLR2-MyD88 pathway (Supplementary Fig. 15b).

The emergence of multi-drug resistant strains and the lack of a vaccines has made *S. aureus* a global concern. Immune evasion has been proven to be the main pathogeneses for *S. aureus* infection[9], and is also one of the main challenges for vaccine development[45]. We identified that K27-linked ubiquitination of Efb by RNF114 binding to TRAF3 disrupts the formation of the TRAF3/TRAF2/cIAP1 complex and prevents K48-ubiquitination-mediated TRAF3 degradation, resulting in suppressed pro-inflammatory cytokine production in vivo and in vitro. The present findings identify a previously unrecognized mechanism that *S. aureus* uses to suppress host immunity (Supplementary Fig. 16). In addition, the Efb-RNF114 interface may be a promising target for the development of effective anti-*S. aureus* treatments.

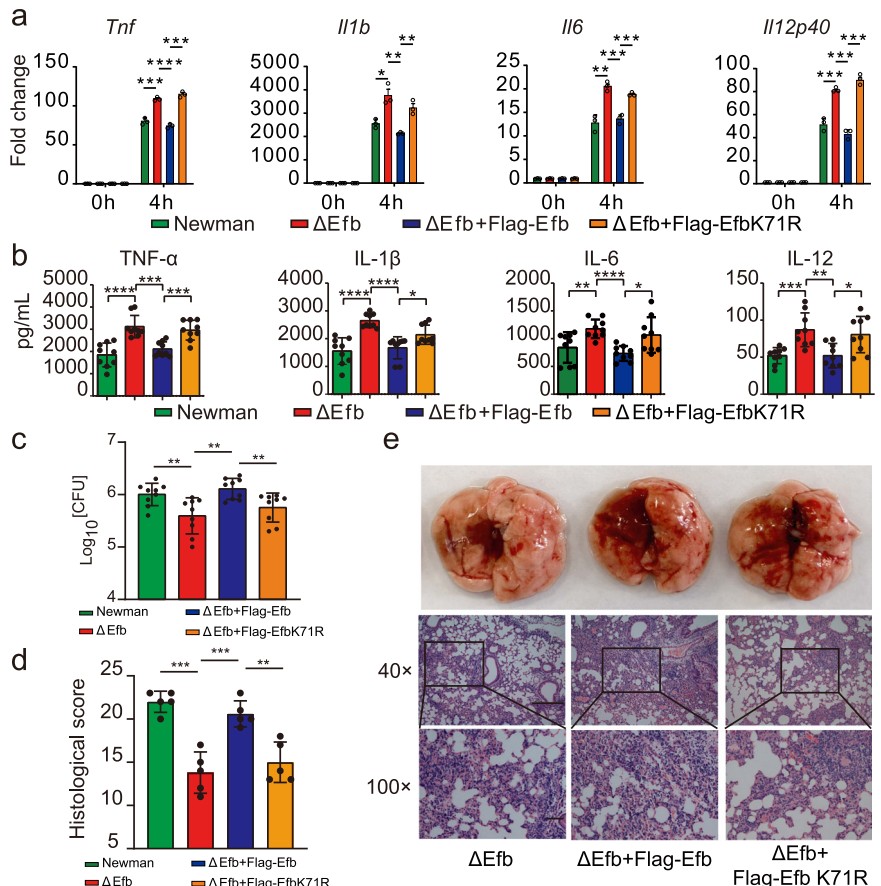

**Fig. 6 | Efb inhibits host immunity depending on K27-ubiquitination. a** qPCR analysis of *Tnf*, *Il1b*, *Il6*, and *Il12p40* mRNA from PMs infected with Newman, ΔEfb, ΔEfb + Flag − Efb, or ΔEfb + Flag − Efb K71R for indicated times (MOI = 2 5; ***P = 0.0001, ****P < 0.0001, ***P = 0.0004, in sequence, *Tnf*; *P = 0.0128, **P = 0.0034, 0.0034, in sequence, *Il1b*; **P = 0.0024, ***P = 0.0010, 0.0010, in sequence, *Il6*; ***P = 0.0010, 0.0001, 0.0002, in sequence, *Il12p40*). **b** ELISA quantification of TNF-α, IL-1β, IL-6, and IL-12 levels in lung tissues homogenized in 1 ml PBS, infected with Newman, ΔEfb, ΔEfb + Flag − Efb, or ΔEfb + Flag − Efb K71R (2 × 10⁸ CFUs per mouse) for 24 h (****P < 0.0001, ***P = 0.0001, 0.0003, in sequence, TNF-α; ****P < 0.0001, <0.0001, *P = 0.0163, in sequence, IL-1β; **P = 0.0065, ****P < 0.0001, *P = 0.0111, in

sequence, IL-6; ***P = 0.0008, **P = 0.0019, *P = 0.0105, in sequence, IL-12). **c** Quantification of the bacterial CFUs in lung tissue homogenates obtained in b (**P = 0.0079, 0.0013, 0.0065). **d, e** Histopathology of lung tissues was assessed in H&E sections stained from mice infected for 24 h; scale bars, 1,000 µm (top) and 200 µm (bottom), the boxed areas at the top are enlarged below (***P = 0.0001, 0.0007, **P = 0.0020). Student's two-tailed unpaired *t*-test (**a**, **b**) or two-tailed Mann−Whitney *U* test (**c**) was used for statistical analysis. Data are representative of three experiments with at least three independent biological replicates. The bars show the mean and standard deviation of n = 3 (a), n = 9 (**b**, **c**), and n = 5 (**d**) per group. Source data are provided as a Source Data file.

## Methods

### Bacterial strains and cells

Bacterial strains adopted in the present study are described in Supplementary Table 1. The *Escherichia coli* strains DH5α and BL21 (Tiangen Biotech, China) were grown in LB medium. When required, the antibiotics ampicillin (100 µg/ml) (Sanggon Biotech) or kanamycin (50 µg/ml) (Sanggon Biotech) were used for the *E. coli* strain selection. The *S. aureus* strains were grown in trypticase soy broth (OXOID, CM0129B). Chloramphenicol (25 µg/ml) (Sanggon Biotech) was used for the selection of *S. aureus* strains when required. Unless noted, all bacteria were grown at 37 °C in a shaking incubator at 200 rpm in tubes kept at a 45° angle.

HEK293T cells (CRL-3216), obtained from the American Type Culture Collection, were maintained in Dulbecco's modified Eagle's medium (DMEM; HyClone) supplemented with 10% (v/v) fetal bovine serum (FBS, HyClone) and 100 U/ml penicillin and streptomycin (HyClone). MH-S cells (CRL-2019), obtained from the American Type Culture Collection, were maintained in Roswell Park Memorial Institute (RPMI)−1640 medium (HyClone) supplemented with 10% FBS. Peritoneal macrophages (PMs) were obtained from 6-week-old wild-type or mutant C57BL/6J mice as follows: mutant mice and their wild-type

littermates were injected with 2 ml Thioglycolate Broth (4%) intraperitoneally. Three days later, the peritoneal lavage fluid was collected from the mice and washed with PBS three times. PMs were grown in DMEM supplemented with 10% FBS. Neutrophils were isolated from mice blood using anti-Ly6G MicroBeads (Miltenyi, 130-120-337) and cultured in RPMI-1640 medium (HyClone) containing 10% FBS.

### Plasmids, reagents, and antibodies

Expressing plasmids were constructed by inserting a synthetic gene segment in the vector, the names of which are listed in Supplementary Table 1. The following antibodies were used for western blot, immunoprecipitation, or immunofluorescence assays: rabbit anti-TRAF3 (PA5-20165, Invitrogen; ab36988, Abcam), mouse anti-TRAF3 (sc6933, Santa Cruz), rabbit anti-TRAF2 (4724, Cell signaling technology, CST), mouse anti-Flag (F1804, Sigma-Aldrich), rabbit anti-HA (3724, CST), rabbit anti-Myc (2040, CST), rabbit anti-phospho-p65 (3033, CST), rabbit anti-phospho-p38 (9215, CST), rabbit anti-phospho-Erk1/2 (9101, CST), rabbit anti-phospho-JNK (4668, CST), rabbit anti-GFP (2956, CST), rabbit anti-K27 (ab181537, Abcam), rabbit anti-K48 (8081, CST), rabbit anti-K63 (5621, CST), rabbit anti-cIAP1 (ab2399, Abcam), rabbit anti-RNF114 (ab97303,

Abcam), Alexa Fluor Plus 488 conjugated goat anti-mouse IgG (A32723, Invitrogen), Alexa Fluor Plus 555 conjugated goat anti-mouse IgG (A32727, Invitrogen), Alexa Fluor Plus 647 conjugated goat anti-rabbit IgG (A32733, Invitrogen); Anti-Efb was generated by immunization of rabbits with the protein of Efb, rabbit anti-GST (CW0085M, Cwbio), rabbit anti-His (CW0083M Cwbio); rabbit anti-GAPDH (G9545, Sigma-Aldrich), mouse anti-Flag M2 Affinity Gel (A2220, Sigma-Aldrich), mouse anti-HA Magnetic Beads (88836, Thermo Fisher), mouse anti-TRAF3 agarose beads (sc6933 AC, Santa Cruz), and goat anti-rabbit IgG (5127, CST), goat anti-mouse IgG (96714, CST). For western blot assays, all primary antibodies were diluted at 1:1000 and secondary antibodies were diluted at a 1:5000. For immunofluorescence, monoclonal mouse anti-Flag M2 antibody was used at 1:500 dilution and rabbit anti-TRAF3 at a 1:200 dilution. Corresponding Alexa Fluor Plus 488 labeled goat anti-mouse IgG, Alexa Fluor Plus 555 labeled goat anti-mouse IgG or Alexa Fluor Plus 647 labeled goat anti-rabbit IgG were used as at a 1:200 dilution. DAPI Stain Solution for nuclear strain was from Sangon Biotech (E607303). For flow cytometry, rat anti-mouse Ly6G PE (551461, BD), rat anti-mouse CD11b-FITC (557396, BD) were used at a 1:100 dilution.

## Construction of *S. aureus* strains
The pBT2 vector (provided by X. Rao, Army Medical University, Chongqing) was used to generate the *S. aureus* Newman strain with a deletion of the gene encoding Efb (ΔEfb) using an allelic replacement strategy as previously described[46]. The primers for the construction of the Efb knockout vector are listed in Supplementary Table 2. The deletion of Efb was confirmed by PCR and Sanger sequencing. The pLI50 vector (provided by X. Rao, Army Medical University, Chongqing) was used to complement the ΔEfb strain with the wild-type Efb or Efb K71R gene driven by the original promotor of Efb. The expression of Efb or its mutants in the supernatant of *S. aureus* was examined by immunoblot assays.

## Transfection and confocal microscopy
HEK293T cells were transiently transfected using PEI (23966-2; Polysciences) according to instructions of the manufacturer. PMs were transduced or transfected with adenovirus (Hanbio) or jetMSSENGER (150-07, Polyplus). MH-S cells were transiently transfected using the INVI DNA RNA Transfection Reagent (IV1216100, Invigentech). Confocal microscopy was performed as described previously[46]. MH-S Cells were fixed with 4% formaldehyde for 10 min at 25 °C, permeabilized for 30 min in PBS containing 0.3% Triton X-100, and then blocked for 1 h at 4 °C in a blocking buffer (1% BSA in PBS). Next, the cells were incubated with the indicated antibodies at 4 °C overnight and secondary antibodies at room temperature for 1 h. After staining with DAPI, images were obtained using a Zeiss LSM 780 confocal laser microscopy system.

## Immunoelectronmicroscopy
After 4 h infection with Newman or ΔEfb strains, MH-S cells were fixed in immunoelectronmicroscopy fixative (Wanwu; G1124) for 2 h at 4 °C. Ultrathin cryosections (70 nm) were prepared as previously described[47], and sections were sequentially labeled with rabbit anti-Flag antibody, followed by sheep anti-rabbit antibody coupled with 10 nm gold particles. The stained sections were observed under a Hitachi electron microscope HT7800.

## Luciferase assay
HEK293T cells were transiently transfected with pNF-κB–luc, pRL–TK, and the indicated plasmids for 24 h. After TNF-α (210-TA, R&D) stimulation for 6 h, the dual-luciferase reporter assay system (RG028, Beyotime) was used to detect luciferase activity according to the instruction of the manufacturer.

## Immunoprecipitation and immunoblot assays
Immunoprecipitation and immunoblot assays were performed as previously described. Briefly, HEK293T cells were transiently transfected with plasmids using PEI. After 48 h, culture supernatants were removed at corresponding times, and the cells were washed three times with PBS. Cells were lysed in cell lysis buffer (Beyotime) supplemented with 1% protease inhibitor cocktail (4693116001, Roche). After centrifugation, the supernatants of cell lysates were incubated with indicated gels at 4 °C overnight. For endogenous immunoprecipitations, PMs and MH-S were infected with *S. aureus* for the indicated times. The cell lysates were subsequently incubated with indicated gels at 4 °C overnight. After being centrifuged, the gels were then washed three times with cell lysis buffer and boiled with 1× SDS loading buffer. Equivalent amounts of total proteins were separated by SDS-PAGE and electro-blotted onto PVDF membranes. The membranes were then probed with antibodies, followed by exposure conducted using an ECL reagent (32209 or 34095, Thermo Fisher).

## Ubiquitination assay
HA-TRAF3 was purified from HEK293T cells using an anti-Flag M2 Affinity Gel. The beads were washed three times with cell lysis buffer. Beads were incubated in E3 ligase buffer, which contained Flag-TRAF2 and Flag-cIAP1 overexpressing cell lysate, ubiquitin, Mg-ATP, with or without E1, E2, or Flag-Efb overexpressing cell lysate. All samples were incubated at 37 °C for 1 h in a metal bath by gently shaking. After removing the supernatants, 50 μl 2× SDS-PAGE gel loading buffer was added to the obtained beads followed by heating to 95 °C for 5 min prior to western blot analysis.

## Real-time PCR analysis
RNA preparation and qPCR analysis were performed as described previously[30] using gene-specific primers (Supplementary Table 2). Total RNA of cells was isolated using RNAiso Plus (9109; Takara). Next, RNA (1 μg) was reverse-transcribed using the PrimeScript™ RT Reagent Kit (RR037; Takara) to generate cDNA. SYBR RT-PCR Kit (QPK-212; Toyobo) were used for the quantitative real-time RT-PCR analysis. Gene amplification was performed using the ΔΔCt method, and gene expression was normalized to that of GAPDH.

## Mice infection model
Six-week-old female specific-pathogen-free C57BL/6 mice were purchased from Beijing HFK Bioscience CO., LTD. Traf3[flox/flox] and Traf3[flox/flox, Lyz2-Cre] mice were purchased from Cyagen Biosciences. They were housed under 12 light/12 dark cycles, ~18–23 °C, 40–60% humidity and specific-pathogen-free (SPF) conditions at the National Engineering Research Center of Immunological Products. All animal experiments were reviewed and approved by the Animal Experiment Administration Committee of Army Medical University and were conducted in accordance with governmental guidelines and institutional policies for the Care and Use of Laboratory Animals. Six-week-old C57BL/6 mice were divided randomly into cages and infected by intratracheal administration with $2 \times 10^8$ CFUs or $6 \times 10^8$ CFUs of different *S. aureus* strains in 20 μl PBS for 24 h or a few days. After the mice were sacrificed at the indicated times, the lungs were collected and homogenized in 1 ml of PBS for ELISA and CFU assays. *S. aureus* burden was determined by plating serial dilutions of each tissue homogenate on tryptic soy agar (TSA) plates, which were incubated at 37 °C. Colonies were counted after 12 h of incubation. For histological analysis, lungs were removed and fixed in 4% paraformaldehyde in PBS and embedded in paraffin. Sections were cut and stained with hematoxylin and eosin (H&E) according to standard protocols. Imaging was performed using microscopy (ECLIPSE 80i, Nikon). For skin and blood infection models, the doses of *S. aureus* were $2 \times 10^8$ and $1 \times 10^8$ CFUs. All mice were age- and sex-matched in each experiment. The sample size was determined based on data from

pilot experiments. For in vitro *S. aureus* infections, macrophages were infected with a single-cell suspension of bacteria at an moieties of infection (MOI) of 25.

## Statistics and reproducibility

Data are expressed as mean ± standard deviation (SD). GraphPad Prism 8 was used for statistical analysis. The sample sizes, reproducibility of experiments and the statistical tests used are presented in the figure legends. ZEN 2.1 on Zeiss LSM 780 confocal laser microscopy system was used for immunohistochemistry data collection and analysis. CFX Manager Software v3.0 on BioRad CFX96Touch was used for qRT-PCR data collection and analysis. Bio-Rad ChemDoc Touch was used for western blot data collection and analysis. BD biosciences FACSDiva software on FACSCanto was used for Flow cytometry data collection and analysis.

## Reporting summary

Further information on research design is available in the Nature Research Reporting Summary linked to this article.

## Data availability

All data are available within the present article and Supplementary Information, or in the Source Data files. Source data are provided with this paper.

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

## Acknowledgements

We thank Xiancai Rao (Third Military Medical University, China) for providing the pBT2 and pLI50 vectors and *S. aureus* Newman strain; Lixin Zheng (LISB/NIAID/NIH, USA) for critical reading of the manuscript; and Weilong Shang (Third Military Medical University, China), Yi Yang (Third Military Medical University, China), and Xianzhi Meng (Southwest University, China) for technical assistance. This work was supported by grants from the National Natural Science Foundation of China (Nos. 81902036, 31970138) and the National Natural Science Foundation of Chongqing (cstc2019jcyj-msxmX0377).

## Author contributions

Q.Z., D.Y., H.Z., and X.Z. designed the experiments. X.Z. and T.X. wrote the manuscript. X.Z., T.X., L.G., Y.S., J.Z., and Z.Z. analyzed experimental results. X.Z., T.X., L.G., Y.W., L.L., and T.T. carried out the experiments. D.L., P.L., W.Z., P.C., H.J., and Q.G. provided technical help. All authors discussed the results and commented on the manuscript.

## Competing interests

The authors declare no competing interests.
