## [Peer review file · Nature Communications]

REVIEWER COMMENTS

Reviewer #1 (Remarks to the Author):

In the manuscript entitled “Efb released by intracellular *Staphylococcus aureus* suppresses host immunity by targeting TRAF3”, the authors studied the role of intracellular *S. aureus* secreted fibrinogen 29 binding protein (Efb) in suppressing proinflammatory cytokine expression and promoting bacterial growth in mice. They found that Efb inhibited NF- κ B activation through disrupting the TRAF3/TRAF2/cIAP1 complex. While the involvement of Efb and RNF114 in regulating TRAF3 stability and inflammatory cytokines are potentially interesting, authors did not provide enough convincing data demonstrating that Efb promotes *S. aureus* growth in mice through inhibiting TRAF3-mediated NF- κ B activation and inflammatory.

Major concerns:

- 1) The rationale for selecting Efb as an NF- κ B inhibitor was not clear. Efb was about the average among 78 secretory proteins of *S. aureus* tested, inhibiting about 50% of NF- κ B activity. Why did author select Efb for their studies and how were other secretory proteins that had stronger effects on NF- κ B inhibition?
- 2) *S. aureus* is classically considered an extracellular pathogen, which can be uptake by macrophages into phagosomes. In order to prove the hypothesis that *S. aureus* protein Efb secreted into macrophage cytoplasm to inhibit NF- κ B, the authors need to distinguish whether Efb is located in the cytoplasm or phagosomes.
- 3) Mice infected with Δ Efb *S. aureus* had about 10 times lower bacterial loads and much higher survival rate than mice infected with either Newman or Δ Efb+Flag-Efb *S. aureus*. However, the differences in the levels of various inflammatory cytokines were about two folds. It seems that Efb may have other unknown functions in addition to suppressing host inflammatory response.
- 4) The authors concluded that Efb inhibits host proinflammatory responses via TRAF3 as there were no significant differences in inflammatory responses of TRAF3 knockout cells infected with Newman and Δ Efb *S. aureus*. However, as shown in Figure 3A, TRAF3 knockout cells induced much higher levels of inflammatory cytokines than the corresponding WT cells, which may mask the differences between Newman and Δ Efb *S. aureus*.

5) In Figure 4, the authors need to provide more direct evidence demonstrating that Efb stabilizes TRAF3 is through preventing cIAP1/2-mediated K48 Ubiquitination and degradation.

6) The major function of the TRAF2, TRAF3 and cIAP1/2 complex is to degrade NIK and regulate the basal levels of the non-canonic NF-kB. The author did not test whether Efb suppresses inflammatory response through inhibiting canonic or non-canonic NF-kB activity.

7) TRAF3 is a K63 ubiquitin ligase, which plays an important role in regulating NF-kB activation. While the authors showed the effects of Efb and RNF114 on inhibiting TRAF3 K48 ubiquitination, they did not show if Efb and RNF114 affect TRAF3 K63 ubiquitination.

8), In the Extended Data Fig. 11, it is not clear why Δ Efb *S. aureus* induced lower levels of inflammatory cytokines than Newman *S. aureus* in RNF114 knockdown cells. Authors should also compare the levels of TRAF3 at different time points after infecting WT and RNF114 knockdown cells with Newman and Δ Efb *S. aureus*.

9) The role of RNF114 in Efb-mediated regulation of both canonic or non-canonic NF-kB activity should also be addressed.

Reviewer #2 (Remarks to the Author):

Comments to manuscript titled "Efb released by intracellular *Staphylococcus aureus* suppresses host immunity by targeting TRAF3".

This manuscript provides strong evidences supporting a novel mechanism exploited by *S. aureus* for avoiding the host's immune response and establish infection. Intercepting this mechanism may represent a potential way for novel therapies against this deadly pathogen.

I have a few, but important points, that the authors may want to address:

1) The authors abundantly stress the role of the inflammatory response dampen by Efb. However, the mechanism that most of the staphylococccologists consider critical for combatting *S. aureus* is opsonophagocytic killing. I think that it is critical that the authors assess if lack of Efb increases the ability of different phagocytes to kill the bacterium.

Minor comments:

1) The authors should explain why they did not consider other staphylococccal factors that in extended Fig. 1 are shown to inhibit NFKb even more than Efb.

2) The authors may want to include NFKb on the model shown in Extended Fig. 12.

Reviewer #4 (Remarks to the Author):

Comment on "Efb released by intracellular 1 *Staphylococcus aureus* suppresses host immunity by targeting TRAF3"

The excreted fibrinogen binding protein, Efb, in *S. aureus* Newman has been described as an immunosuppressive, immune evasive, and complement inhibitory protein. Here, it is reported that Efb is secreted into the cytoplasm after phagocytosis of SA Newman by macrophages, where it is inhibiting the expression of proinflammatory cytokines (TNF- α , IL-1 β , IL-6 and IL-12). The uncovering of the underlying mechanism is the core of this work. Apparently, a cascade of reactions leads to suppression host immunity. RING finger protein 114, a host E3 ligase, ubiquitinates of Efb at lysine 71, which facilitates the recruitment of TNF receptor associated factor 3 (TRAF3). The binding of Efb to TRAF3 disrupts the formation of the TRAF3/TRAF2/cIAP1 complex, which mediates K48-ubiquitination of TRAF3 to promote degradation, resulting in suppression of the inflammatory signaling cascade. The importance of ubiquitination of Efb at lysine 71 was confirmed by Efb K71R mutant, which lost suppression of proinflammatory cytokines.

The authors have performed an exceptional large number of experiments to document the mechanistic relationships of the underlying immunosuppression of Efb. The data analysis, interpretation and conclusions are as far as I can judge conclusive, only the role postulated role of TLR2 is unclear.

There are some questions and a recommendations:

Extended Data Fig. 12. It would be good to explain the reaction cascade in more detail. Is there a cascade at all and what is the first reaction and the following reactions?

The role of TLR2 unclear. It is known that TLR2 priming of macrophages results in increased TRAF3 levels that lead to the selective priming of IFN- β production by multiple innate immune. To substantiate the role of TLR2 it would be necessary to also investigate a lgt mutant of Newman (Schmaler et al., 2009). One cannot rule out that some of the described effects are mediated by lipoprotein-induced TLR2 activation, even though the Efb K71R mutant suggests that Efb is a main driver.

Why using Flag-Efb? Is the Anti-Flag Antibody not interfering with Spa?

Is TLR2 directly connected with TRAF3 or with Myd88?

In how many clinical isolates of *S. aureus* express Efb?

**Author responses to Reviewer #1**

In the manuscript entitled “Efb released by intracellular *Staphylococcus aureus* suppresses host
immunity by targeting TRAF3”, the authors studied the role of intracellular *S. aureus* secreted
fibrinogen 29 binding protein (Efb) in suppressing proinflammatory cytokine expression and
promoting bacterial growth in mice. They found that Efb inhibited NF- κ B activation through
disrupting the TRAF3/TRAF2/cIAP1 complex. While the involvement of Efb and RNF114 in
regulating TRAF3 stability and inflammatory cytokines are potentially interesting, authors did not
provide enough convincing data demonstrating that Efb promotes *S. aureus* growth in mice through
inhibiting TRAF3-mediated NF- κ B activation and inflammatory.

**Response:** We thank the reviewer for the comments and constructive suggestions on our paper.

Major concerns:

1) The rationale for selecting Efb as an NF- κ B inhibitor was not clear. Efb was about the average
among 78 secretory proteins of *S. aureus* tested, inhibiting about 50% of NF- κ B activity. Why
did author select Efb for their studies and how were other secretory proteins that had stronger
effects on NF- κ B inhibition?

**Response:** Thank you for the feedback. We have now explained why we chose Efb in lines 295-311
of the revised manuscript. Here, we also would like to provide some verified results about
macrophages infected with wild type and corresponding knockout *S. aureus* strains. These results
indicate that these three proteins are either not produced or have no effect on macrophage functions
during *S. aureus* infection. The validation work is still in progress.

2) *S. aureus* is classically considered an extracellular pathogen, which can be uptake by

macrophages into phagosomes. In order to prove the hypothesis that *S. aureus* protein Efb
secreted into macrophage cytoplasm to inhibit NF- κ B, the authors need to distinguish whether
Efb is located in the cytoplasm or phagosomes.

**Response:** As per the reviewer's suggestion, we performed an immunoelectron microscopy assay.
The method descriptions were added in lines 451-457 of the revised manuscript. The results show
that Efb exists in the cytoplasm of macrophages. We supplied our results in the following figure and
added representative results in Extended Data Fig. 6a of the revised manuscript.

3) Mice infected with ΔEfb *S. aureus* had about 10 times lower bacterial loads and much higher
survival rate than mice infected with either Newman or ΔEfb+Flag-Efb *S. aureus*. However, the
differences in the levels of various inflammatory cytokines were about two folds. It seems that
Efb may have other unknown functions in addition to suppressing host inflammatory response.

**Response:** Thank you for raising this point. Figure 6 a, b indicates that, when compared to wild-
type Efb, mutant Efb (EfbK71R), which does not influence the effect on C3b and fibrinogen (lines
333-344 of the revised manuscript), loses its inhibition effects on host inflammatory responses and

induces less *S. aureus* loads and lower pathogenicity in mice. Therefore, we believe that the anti-
inflammatory effects of Efb are critical to *S. aureus* load and pathogenicity changes in mice. We
have also added some illustrations in lines 346-349 of the revised manuscript for clarity.

4) The authors concluded that Efb inhibits host proinflammatory responses via TRAF3 as there
were no significant differences in inflammatory responses of TRAF3 knockout cells infected
with Newman and Δ Efb *S. aureus*. However, as shown in Figure 3A, TRAF3 knockout cells
induced much higher levels of inflammatory cytokines than the corresponding WT cells, which
may mask the differences between Newman and Δ Efb *S. aureus*.

**Response:** Thank you for the feedback. We performed a new experiment to supply more supporting
data (see line 206-211 and Extended Data Fig. 9 i of the revised manuscript). We think the result
and figure 3a suggest that Efb inhibits host proinflammatory responses is mainly via TRAF3.

5) In Figure 4, the authors need to provide more direct evidence demonstrating that Efb stabilizes
TRAF3 is through preventing cIAP1/2-mediated K48 Ubiquitination and degradation.

**Response:** As suggested by the reviewer, we began by overexpressing TRAF3, TRAF2, cIAP1, Efb,
and K48 Ub in corresponding HEK 293T cells. The results suggest that Efb could inhibit K48-linked
ubiquitination of TRAF3 in the presence of TRAF2 and cIAP1 (Extended Data Fig. 9f of the revised
manuscript). Next, we repeated this experiment in a cell free ubiquitylation reaction system and
obtained a similar result (Fig. 4f of the revised manuscript). The results descriptions were added in
lines 198-200 of the revised manuscript, and method descriptions were added in lines 480-487.

6) The major function of the TRAF2, TRAF3 and cIAP1/2 complex is to degrade NIK and regulate
the basal levels of the non-canonic NF-kB. The author did not test whether Efb suppresses
inflammatory response through inhibiting canonic or non-canonic NF-kB activity.

**Response:** Thank you for the feedback. In addition to mediating the non-canonical NF-kB pathway,
TRAF3 has been shown to be a regulator of the canonical NF-kB, MAPK, and type I interferon
pathways¹. Therefore, we performed new experiments as described in lines 172-183 of the revised
manuscript. Extended Data Fig. 8 shows that Efb suppresses inflammatory responses mainly
through inhibiting the canonical NF-kB and MAPK pathways.

7) TRAF3 is a K63 ubiquitin ligase, which plays an important role in regulating NF-kB activation.
While the authors showed the effects of Efb and RNF114 on inhibiting TRAF3 K48
ubiquitination, they did not show if Efb and RNF114 affect TRAF3 K63 ubiquitination.

**Response:** Thank you for the comment. As reported previously, K63 ubiquitination of TRAF3 can
mediate type I interferon and NF-kB pathways^{2,3}. However, we found no detectable K63
ubiquitination of TRAF3 during infection with either wild type or Δ Efb *S. aureus* (lines 190-193
and Extended Data Fig. 9c of the revised manuscript).

8) In the Extended Data Fig. 11, it is not clear why Δ Efb *S. aureus* induced lower levels of

inflammatory cytokines than Newman *S. aureus* in RNF114 knockdown cells. Authors should
 also compare the levels of TRAF3 at different time points after infecting WT and RNF114
 knockdown cells with Newman and Δ Efb *S. aureus*.

 **Response:** Thank you for the suggestion. Recently we obtained RNF114 knockout mice. As per the
 reviewer's suggestion, we compared the expression levels of TRAF3 in WT and RNF114-/
 macrophages (lines 263-264 and Extended Data Fig. 13g). The results showed that RNF114 has an
 effect on TRAF3 expression only when Efb is present. We also performed the experiment of
 Extended Data Fig. 13f in RNF114-/- macrophages.

 According to the results of two independent biological replicates, we think RNF114 may involve in
 the regulation of inflammatory cytokine production during *S. aureus* infection independent of Efb
 and TRAF3. Next, we will find whether RNF114 mediated other signaling during *S. aureus* infection
 in the future.

 9) The role of RNF114 in Efb-mediated regulation of both canonic or non-canonic NF-kB activity
 should also be addressed.

 **Response:** Thank you for the comment. As we response to comment 6 and 8, Efb suppresses
 inflammatory responses mainly through inhibiting canonic NF-kB and MAPK pathways. And our
 main conclusion is that RNF114 facilitate Efb to inhibit canonic NF-kB and MAPK pathways,
 resulting in inhibition effects of host proinflammatory responses. Next, we will further explore the
 effects of RNF114 in *S. aureus* infection.

**References:**

1 Tseng, P. H. et al., Different modes of ubiquitination of the adaptor TRAF3 selectively activate
 the expression of type I interferons and proinflammatory cytokines. NAT IMMUNOL 11 70 (2010).

2 Zhu, Q. et al., TRIM24 facilitates antiviral immunity through mediating K63-linked TRAF3
 ubiquitination. J EXP MED 217 (2020).

3 Zhou, W. et al., Hypothermic oxygenated perfusion inhibits HECTD3-mediated TRAF3
polyubiquitination to alleviate DCD liver ischemia-reperfusion injury. CELL DEATH DIS 12 211
(2021).

**Author responses to Reviewer #2**

Comments to manuscript titled "Efb released by intracellular *Staphylococcus aureus* suppresses host
immunity by targeting TRAF3".

This manuscript provides strong evidences supporting a novel mechanism exploited by *S. aureus*
for avoiding the host's immune response and establish infection. Intercepting this mechanism may
represent a potential way for novel therapies against this deadly pathogen.

**Response:** We appreciate the reviewer's insightful comments and recognition of our work.

I have a few, but important points, that the authors may want to address:

1) The authors abundantly stress the role of the inflammatory response dampen by Efb. However,
the mechanism that most of the staphylococcologists consider critical for combatting *S. aureus* is
opsonophagocytic killing. I think that it is critical that the authors assess if lack of Efb increases the
ability of different phagocytes to kill the bacterium.

**Response:** Thank you for raising this point. Opsonophagocytic killing conducted by macrophages
and neutrophils is critical for the host to combat *S. aureus*. Per the reviewer's suggestion, we recently
tested this mechanism (lines 288-291, 311-313 and Extended Data Fig. 14 of the revised manuscript)
and found that Efb did not affect *S. aureus* survival when cocultured with isolated neutrophils.
Therefore, we believe inflammatory responses of macrophages dampened by Efb may lead to
reduced chemotaxis or activation of neutrophils or other immune cells, resulting in enhanced *S.*
*aureus* survival *in vivo*.

Minor comments:

1) The authors should explain why they did not consider other staphylococcal factors that in
extended Fig. 1 are shown to inhibit NFKb even more than Efb.

**Response:** Thank you for the feedback. We have now explained why we chose Efb in lines 295-311
of the revised manuscript. Here, we also would like to provide some verified results about
macrophages infected with wild type and corresponding knockout *S. aureus* strains. These results
indicate that these three proteins are either not produced or have no effect on macrophage functions
during *S. aureus* infection. The validation work is still in progress.

2) The authors may want to include NFKb on the model shown in Extended Fig. 12.

**Response:** We revised Extended Fig. 12 (Extended Fig. 16 of the revised manuscript) as the
 reviewer suggested.

**Author's response to Reviewer #4**

Comment on "Efb released by intracellular 1 *Staphylococcus aureus* suppresses host immunity by
targeting TRAF3"

The excreted fibrinogen binding protein, Efb, in *S. aureus* Newman has been described as an
immunosuppressive, immune evasive, and complement inhibitory protein. Here, it is reported that
Efb is secreted into the cytoplasm after phagocytosis of SA Newman by macrophages, where it is
inhibiting the expression of proinflammatory cytokines (TNF- α , IL-1 β , IL-6 and IL-12). The
uncovering of the underlying mechanism is the core of this work. Apparently, a cascade of reactions
leads to suppression host immunity. RING finger protein 114, a host E3 ligase, ubiquitinates of Efb
at lysine 71, which facilitates the recruitment of TNF receptor associated factor 3 (TRAF3). The
binding of Efb to TRAF3 disrupts the formation of the TRAF3/TRAF2/cIAP1 complex, which
mediates K48-ubiquitination of TRAF3 to promote degradation, resulting in suppression of the
inflammatory signaling cascade. The importance of ubiquitination of Efb at lysine 71 was confirmed
by Efb K71R mutant, which lost suppression of proinflammatory cytokines.

The authors have performed an exceptional large number of experiments to document the
mechanistic relationships of the underlying immunosuppression of Efb. The data analysis,
interpretation and conclusions are as far as I can judge conclusive, only the role postulated role of
TLR2 is unclear.

There are some questions and a recommendations:

**Response:** We appreciate the reviewer's insightful comments and recognition of our work.

Extended Data Fig. 12. It would be good to explain the reaction cascade in more detail. Is there a
cascade at all and what is the first reaction and the following reactions?

**Response:** Thank you for the suggestion. As stated previously, TLR2 of innate immune cells plays
a key role in detecting *S. aureus*^{1,2}. We confirmed that TLR2, MyD88, and TAK1 mainly mediated
pro-inflammatory cytokine production during *S. aureus* infection (see lines 172-183, 362-371 and
Extended Data Fig. 8 and 15 of the revised manuscript). We have added some information regarding
these data in Extended Data Fig. 16.

The role of TLR2 unclear. It is known that TLR2 priming of macrophages results in increased
TRAF3 levels that lead to the selective priming of IFN- β production by multiple innate immune. To
substantiate the role of TLR2 it would be necessary to also investigate a lgt mutant of Newman
(Schmaler et al., 2009). One cannot rule out that some of the described effects are mediated by
lipoprotein-induced TLR2 activation, even though the Efb K71R mutant suggests that Efb is a main
driver.

**Response:** Thank you for the suggestion. We agree with the reviewer that the role of TLR2 in our
paper is unclear. As stated in a report by Perkins et al,³ macrophages stimulated by P3C, a TLR2

ligand, overnight (≥ 24 h) upregulate TRAF3 protein expression. However, the TRAF3 mRNA level
was not significantly increased until 8 hours. The effects on TRAF3 protein expression by
immediate TLR2 activation have not yet been explored.

We measured TRAF3 protein expression immediately after *S. aureus* infection. We also think using
lgt mutant of Newman, which still has other TLR2 ligands, such as LTA (lipoteichoic acid), to
determine the role of TLR2 is an indirect means of measuring changes in TRAF3 expression.
Therefore, we performed the experiment on TLR2 and MyD88 knockout macrophages (see lines
362-371 and Extended Data Fig. 15 of the revised manuscript). We posit that a decrease in TRAF3
protein expression is related to TLR2-MyD88 signaling.

Why using Flag-Efb? Is the Anti-Flag Antibody not interfering with Spa?

**Response:** Thank you for raising this question. There are currently no anti-Efb antibodies on the
market. We prepared a polyclonal anti-Efb antibody from rabbit. However, it cannot be used in
immunofluorescence experiments. In the extended Fig. 2b of the revised manuscript, we can see
that the anti-Flag-Efb we used does not bind wild type Neman, which suggests this antibody does
not interfere with Spa.

Is TLR2 directly connected with TRAF3 or with Myd88?

**Response:** Thank you for raising this question. As stated in a previous report, Myd88, but not TLR2,
may directly interacts with TRAF3⁴.

In how many clinical isolates of *S. aureus* express Efb?

**Response:** Thank you for raising this question. As stated in a previous report, all clinical isolates of
*S. aureus* express highly conserved Efb⁵. According to our BLAST alignment analysis, the Efb
coding sequence is present in 99% of genomes of *S. aureus* strains (1,002 strains with completed
genomes) in the NCBI database, as listed in the supplementary information.

**References:**

1 Schmalzer, M. et al., Lipoproteins in *Staphylococcus aureus* mediate inflammation by TLR2

and iron-dependent growth in vivo. *J IMMUNOL* 182 7110 (2009).
2 Wang, X., Eagen, W. J. & Lee, J. C., Orchestration of human macrophage NLRP3
inflammasome activation by *Staphylococcus aureus* extracellular vesicles. *Proc Natl Acad Sci U S*
*A* 117 3174 (2020).
3 Perkins, D. J. et al., Reprogramming of murine macrophages through TLR2 confers viral
resistance via TRAF3-mediated, enhanced interferon production. *PLOS PATHOG* 9 e1003479
(2013).
4 Lalani, A. I., Luo, C., Han, Y. & Xie, P., TRAF3: a novel tumor suppressor gene in macrophages.
*Macrophage (Houst)* 2 e1009 (2015).
5 Boden, W. M. & Flock, J. I., Incidence of the highly conserved fib gene and expression of the
fibrinogen-binding (Fib) protein among clinical isolates of *Staphylococcus aureus*. *J CLIN*
*MICROBIOL* 33 2347 (1995).

REVIEWERS' COMMENTS:

Reviewer #1 (Remarks to the Author):

The revised manuscript has addressed most of the concerns I raised during the initial review.

Reviewer #3 (Remarks to the Author):

In the revised version, all my comments and suggestions have been taken into account to my fullest satisfaction.

However, I still have a small comment on the authors' comment, but it does not require any postponement of a possible acceptance of the work.

277 We measured TRAF3 protein expression immediately after *S. aureus* infection. We also think using lgt mutant of Newman, which still has other TLR2 ligands, such as LTA (lipoteichoic acid), to determine the role of TLR2 is an indirect means.....

It has been described in various papers that lipoteichoic acid (LTA) from *S. aureus* has no significant immunostimulatory effect and does not act as a TLR2 agonist (Hashimoto et al., 2006b, Hashimoto et al., 2006a, Stoll et al., 2005). In the many papers describing TLR2 activation by LTA, the so called "highly purified" LTA was still contaminated with lipopeptides which ultimately triggered the immune stimulation. Zähringer, an expert in lipid and glycolipid structures wrote: "Our data together with those reported by other groups reveal that only lipoproteins/lipopeptides are sensed at physiologically concentrations by TLR2 at picomolar levels. This finding implies that the activity of all other putative bacterial compounds so far reported as TLR2 agonists was most likely due to contaminating highly active natural lipoproteins and/or lipopeptides" (Zähringer et al., 2008).

Hashimoto, M., Tawaratsumida, K., Kariya, H., Aoyama, K., Tamura, T., and Suda, Y. (2006a) Lipoprotein is a predominant Toll-like receptor 2 ligand in *Staphylococcus aureus* cell wall components. *Int Immunol* 18: 355-362.

Hashimoto, M., Tawaratsumida, K., Kariya, H., Kiyohara, A., Suda, Y., Krikae, F., Kirikae, T., and Götz, F. (2006b) Not lipoteichoic acid but lipoproteins appear to be the dominant immunobiologically active compounds in *Staphylococcus aureus*. *J Immunol* 177: 3162-3169.

Stoll, H., Dengjel, J., Nerz, C., and Götz, F. (2005) *Staphylococcus aureus* deficient in lipidation of prelipoproteins is attenuated in growth and immune activation. *Infect Immun* 73: 2411-2423.

Zähringer, U., Lindner, B., Inamura, S., Heine, H., and Alexander, C. (2008) TLR2 - promiscuous or specific? A critical re-evaluation of a receptor expressing apparent broad specificity. *Immunobiology* 213: 205-224.

Reviewer #1 (Remarks to the Author):

The revised manuscript has addressed most of the concerns I raised during the initial review.

Response: We appreciate the reviewer's recognition of our work.

Reviewer #3 (Remarks to the Author):

In the revised version, all my comments and suggestions have been taken into account to my fullest satisfaction.

However, I still have a small comment on the authors' comment, but it does not require any postponement of a possible acceptance of the work.

Response: We appreciate the reviewer's insightful comments and recognition of our work.

277 We measured TRAF3 protein expression immediately after *S. aureus* infection. We also think using Igt mutant of Newman, which still has other TLR2 ligands, such as LTA (lipoteichoic acid), to determine the role of TLR2 is an indirect means.....

It has been described in various papers that lipoteichoic acid (LTA) from *S. aureus* has no significant immunostimulatory effect and does not act as a TLR2 agonist (Hashimoto et al., 2006b, Hashimoto et al., 2006a, Stoll et al., 2005). In the many papers describing TLR2 activation by LTA, the so called "highly purified" LTA was still contaminated with lipopeptides which ultimately triggered the immune stimulation. Zähringer, an expert in lipid and glycolipid structures wrote: "Our data together with those reported by other groups reveal that only lipoproteins/lipopeptides are sensed at physiologically concentrations by TLR2 at picomolar levels. This finding implies that the activity of all other putative bacterial compounds so far reported as TLR2 agonists was most likely due to contaminating highly active natural lipoproteins and/or lipopeptides" (Zähringer et al., 2008).

Hashimoto, M., Tawaratsumida, K., Kariya, H., Aoyama, K., Tamura, T., and Suda, Y. (2006a) Lipoprotein is a predominant Toll-like receptor 2 ligand in *Staphylococcus aureus* cell wall components. *Int Immunol* 18: 355-362.

Hashimoto, M., Tawaratsumida, K., Kariya, H., Kiyohara, A., Suda, Y., Krikae, F., Kirikae, T., and Götz, F. (2006b) Not lipoteichoic acid but lipoproteins appear to be the dominant immunobiologically active compounds in *Staphylococcus aureus*. *J Immunol* 177: 3162-3169.

Stoll, H., Dengjel, J., Nerz, C., and Götz, F. (2005) *Staphylococcus aureus* deficient in lipidation of prelipoproteins is attenuated in growth and immune activation. *Infect Immun* 73: 2411-2423.

Zähringer, U., Lindner, B., Inamura, S., Heine, H., and Alexander, C. (2008) TLR2 - promiscuous or specific? A critical re-evaluation of a receptor expressing apparent broad specificity. *Immunobiology* 213: 205-224.

Response: Thank you for the feedback. We agree with these comments. We hope to use this information as a direction for future studies in order to draw a more convincing conclusion regarding the possible role of TLR2.